# Effect of the Molybdenum Substrate Shape on Mo₂C Coating Electrodeposition

**Anton Dubrovskiy** **, Olga Makarova and Sergey Kuznetsov ***

Tananaev Institute of Chemistry of the Federal Research Centre "Kola Science Centre of the Russian Academy of Sciences", Apatity 184209, Russia; a.dubrovskiy@chemy.kolasc.net.ru (A.D.); makarova@chemy.kolasc.net.ru (O.M.)

* Correspondence: kuznet@chemy.kolasc.net.ru; Tel.: +7-815-557-9730

**Abstract:** By electrochemical synthesis in the NaCl-KCl-Li₂CO₃ (1.5 wt.%)-Na₂MoO₄ (8.0 wt.%) melt on molybdenum, substrates with different configuration Mo₂C coatings with the hexagonal lattice were obtained. The influence of the substrate form on the structure of Mo₂C cathodic deposits was studied. The molybdenum carbide coatings on a molybdenum substrate (Mo₂C/Mo) show a catalytic activity in the water–gas shift (WGS) reaction by at least three orders of magnitude higher than that of the bulk Mo₂C phase. The catalytic activity remained constant during 500 h for the water–gas shift reaction.

**Keywords:** molten salts; different shape substrates; Mo₂C electrodeposition; microstructure of coatings

## 1. Introduction

Transition metal carbides are widely used in metallurgical processes due to their unique refractory properties. Molybdenum carbide (β-Mo₂C) stands out among the group of transition metal carbides because of its unique catalytic properties similar to those of noble metals for a variety of reactions such as Fisher–Tropsch [1], reforming [2,3], water–gas shift [4,5], hydrogenation [6,7], hydrodeoxygenation [8], and CO₂ reduction [9].

Industrially, solid-state reduction between graphitic carbon and molybdenum oxides is commonly employed to produce molybdenum carbide. However, this carbothermic reaction involves very high temperatures (>1273 K), leading to a product with a low specific surface area, which is detrimental for catalytic processes.

Recently, the main method of bulk molybdenum semi-carbide synthesis has been the temperature-programmed reaction (TPR). In the works of different authors [10–12], there are various conditions for the implementation of TPR. The main varied conditions are the rate of temperature change, the source of carbon and its relationship with hydrogen. Basically, a change in these parameters affects the time and temperature of the synthesis, and not the crystalline structure of the semi-carbide; therefore, there is no need for a detailed analysis of various options for TPR.

Historically, the molybdenum carbide coatings on metal substrates can be obtained by vacuum-plasma deposition; by heating a molybdenum substrate in CO atmosphere at 1003 K followed by an annealing step at 1973 K [13]; by reduction of Mo(CO)₆ with hydrogen on a metal wire at 573–1073 K [14]; and by oxidation of ammonium paramolybdate to molybdenum trioxide followed by its reduction in an equimolar mixture of methane and hydrogen [4]. However, the aforementioned methods could not produce Mo₂C with the hexagonal lattice only and contain up to 10 wt.% cubic Mo₂C that decrease drastically the catalytic properties.

The electrodeposition of molybdenum carbide coatings using molten salts [15–20] has the following advantages in comparison to other methods:

- melt electrolysis using pulse and reverse current provides the possibility to easily control the deposit structure, thickness, porosity, degree of roughness, texture of galvanic coatings, and grain size (up to nanosized values);
- the laboratory electrodeposition parameters can be used for the large-scale production;
- uniform coatings can be obtained on substrates of complex shapes;
- production a high purity coatings, even with initial reagents of poor quality;
- low operating costs and a low cost of the electrochemical equipment.

The objective of the present work was the electrodeposition of molybdenum carbide coatings on different shaped molybdenum substrates in molten salts, and its lattice, and microstructure characterization.

This work also presents the development of new generation of highly active and stable catalysts in the form of coatings based on the $Mo_2C/Mo$ system obtained by electrochemical synthesis.

## 2. Materials and Methods

### 2.1. Chemicals, Preparation of Salts

Electrochemical synthesis of molybdenum carbide coatings on a molybdenum support was performed by galvanostatic electrolysis using the $NaCl$-$KCl$-$Li_2CO_3$-$Na_2MoO_4$ molten salt system. Alkali metal chlorides (Prolabo, VWR Singapore Ltd, Singapore, 99.5 wt.%) were recrystallized by prolonged and gradual heating above the melting point in a gaseous HCl atmosphere in quartz ampules. Excess HCl was removed from the melt with an argon flow. Prior to starting the experiments, the salts were stored in a glove box in sealed glass ampules. $Li_2CO_3$ (Sigma-Aldrich, St. Louis, MO, USA, purum, $\geq$99 wt.%) and $Na_2MoO_4 \cdot H_2O$ (Sigma-Aldrich, purum, $\geq$99.5 wt.%) were dried for 24 h at 473 K. Alkali metal chlorides were mixed in the required ratio and charged into a glassy carbon ampule (SU-2000), after which the ampule was placed in a stainless steel retort. The retort was evacuated to a residual pressure of 0.66 Pa first at room temperature and then at 473, 673, and 873 K. The cell was heated using a programmable furnace equipped with a Termodat (Termodat Control Systems Co., Ltd, Perm, Russia) 17E3 temperature regulator. The temperature was measured with a Pt-Rh thermocouple (10 wt.% Pt). The retort was filled with high-purity argon (impurity content: <3 ppm $H_2O$, <2 ppm $O_2$), and the electrolyte was melted. $Li_2CO_3$ and $Na_2MoO_4$ were added to the alkali metal chloride melt.

### 2.2. Molybdenum Substrates

Various molybdenum substrates were used: flat plates (Sigma-Aldrich, $\geq$99.9 wt.%, CAS number 7439-98-7) of size 100 mm × 10 mm × 0.1 mm; goffered plates of the same size with the corrugation height of 1.4 mm; wire (Sigma-Aldrich, wire reel, 50 m, diameter 0.25 mm, annealed, 99.95%, CAS number 7439-98-7) 250 μm in diameter. In order to remove the organic impurities from the surface, the molybdenum plates were placed in boiling xylene for 1 h and were then heated in a furnace at 413 K to desorb the xylene that remained on the surface. Synthesis of $Mo_2C$ on molybdenum plates and wire was performed at 1123 K for 7 h at the cathodic current density of 5 mA·cm$^{-2}$. An ampule made of SU-2000 glassy carbon was used as anode. The samples after the experiment were washed with distilled water and alcohol.

### 2.3. Electrochemical Studies

Electrochemical studies were performed by cyclic voltammetry with linear potential sweep using a VoltaLab-40 potentiostat with VoltaMaster-4 software (version 6). The potential sweep rate was varied from $5 \times 10^{-3}$ to 2.0 V·s$^{-1}$. Experiments were performed in the temperature interval

973–1123 K. Cyclic voltammograms were recorded on molybdenum working electrode 0.5 mm in diameter vs. platinum wire acting as quasi-reference electrode [21]. A glassy carbon crucible acted as an auxiliary electrode.

## 2.4. Coatings Characterization

The cathodic products were identified with a Shimadzu XRD-6000 (Shimadzu, Kyoto, Japan) diffractometer using monochromated Cu K$\alpha$ radiation at the recording rate of 0.25°·min$^{-1}$.

The microstructure was studied using LEO-420 and LEO-1450 scanning electron microscopes and an Observer D1m optical microscope (Carl Zeiss Microscopy, LLC, Thornwood, NY, USA) with Thixomet image analyzer (Thixomet, St.-Petersburg, Russia). Roughness was measured by a Nano-R atomic force microscope (AFM) (Pacific Nanotechnology, Santa Clara, CA, USA) and profilometer 130 (Proton-MIET, Zelenograd, Russia). Measurements on both sides in the longitudinal and transverse direction were carried out for each sample.

## 3. Results and Discussion

### 3.1. Electrochemical Synthesis of $Mo_2C$

Taking into account specific features of the electrodeposition of molybdenum and carbon, we chose for the electrochemical synthesis chloride-carbonate-molybdate melt.

The voltammogram of the NaCl-KCl-Na$_2$MoO$_4$ melt, recorded with the molybdenum cathode, is shown in Figure 1a. It has the same form as the NaCl–KCl melt, since MoO$_4{}^{2-}$ ions are characterized by a more negative discharge potential than Na$^+$ and K$^+$ cations, and they are electrochemically inactive in an equimolar mixture of sodium and potassium chlorides [22].

The voltammogram of the NaCl-KCl-Li$_2$CO$_3$-Na$_2$MoO$_4$ melt, recorded with the molybdenum cathode, is shown in Figure 1b.

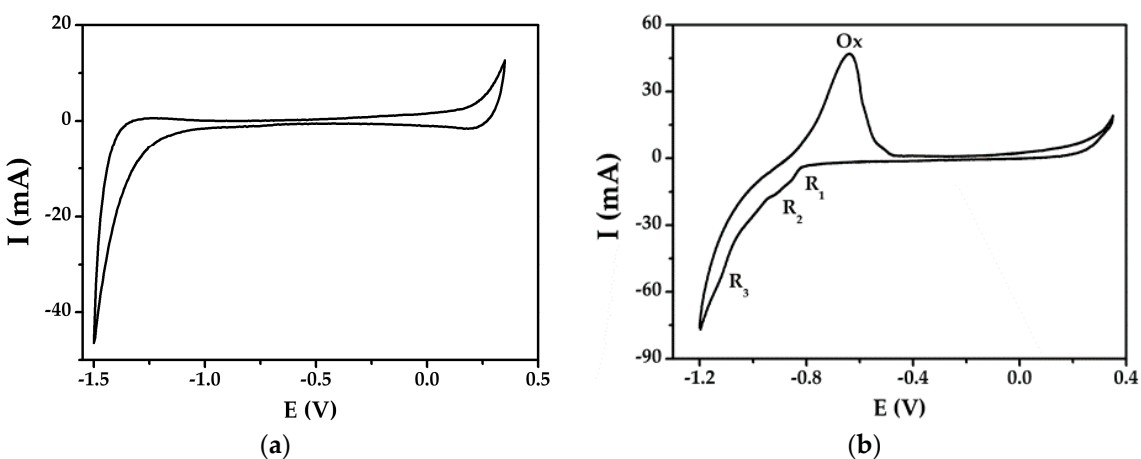

**Figure 1.** Cyclic voltammograms recorded on a molybdenum electrode in the melts: (**a**) NaCl-KCl-Na$_2$MoO$_4$ ($C_{Na_2MoO_4}$ = 3.58 × 10$^{-4}$ mol·cm$^{-3}$), and (**b**) NaCl-KCl-Li$_2$CO$_3$-Na$_2$MoO$_4$ ($C_{Li_2CO_3}$ = 2.8 × 10$^{-4}$ mol·cm$^{-3}$, $C_{Na_2MoO_4}$ = 5.47 × 10$^{-4}$ mol·cm$^{-3}$). Electrode surface area 0.238 cm$^2$, potential sweep rate 0.1 V·s$^{-1}$, temperature 1023 K, platinum quasi-reference electrode.

In the cathodic half-cycle, it has three waves $R_1$, $R_2$, and $R_3$, and in the anodic half-cycle, there is only one O$x$ wave. Potentiostatic electrolysis at potentials of the $R_1$, $R_2$, and $R_3$ waves led, according to the results of X-ray diffraction (XRD) analysis, to the formation of Mo$_2$C. The single wave (O$x$) at anodic polarization confirms the formation of only one product in the course of electrolysis. The cathodic current density for the process in the region of potentials of the $R_1$ wave is low, which is due to low concentration of electroactive carbon-containing species. The $R_1$ wave corresponds to the discharge of

$CO_2$ and is described by electrode Reaction (1) complicated by the preceding chemical Reaction (2). The $R_2$ wave corresponds to the discharge of carbonate ions according to Reaction (3).

$$CO_2 + 4e^- + 2Mo \rightarrow Mo_2C + 2O^{2-} \tag{1}$$

$$CO_3{}^{2-} \leftrightarrow CO_2 + O^{2-} \tag{2}$$

$$CO_3{}^{2-} + 4e^- + 2Mo \rightarrow Mo_2C + 3O^{2-} \tag{3}$$

The electrode process at joint electroreduction of $CO_3{}^{2-}$ and $MoO_4{}^{2-}$ with the formation of $Mo_2C$ can be described in the general form as follows:

$$2MoO_4{}^{2-} + CO_3{}^{2-} + 16e^- \rightarrow Mo_2C + 11O^{2-} \tag{4}$$

Note that the discharge of $MoO_4{}^{2-}$ ions becomes possible due to the appearance of lithium cations in the melt, which shift the potential of the discharge of molybdate ions to the positive region. The use of different electrolysis regimes led to the formation of a single product at the cathode-molybdenum semi-carbide ($Mo_2C$).

### 3.2. X-Ray Diffraction (XRD) Analysis

In all cases, molybdenum semi-carbide with a hexagonal crystal lattice (JCPDS card No. 35-0787) from the NaCl-KCl-Li$_2$CO$_3$ (1.5 wt.%)-Na$_2$MoO$_4$ (8.0 wt.%) melt was obtained (see Figure 2). The formation of $Mo_2C$ with a hexagonal lattice in the process of electrochemical synthesis occurs due to specific conditions (electric field, double layer and high temperature) of the electrocrystallization process.

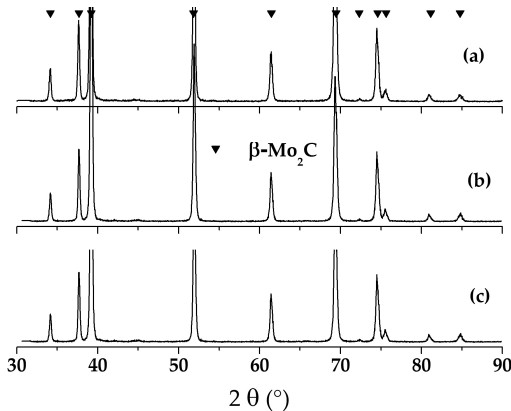

**Figure 2.** X-ray diffraction (XRD) patterns of molybdenum carbide on molybdenum substrates with different shapes: (**a**) straight plate, (**b**) goffered plate, (**c**) wire. NaCl-KCl-Li$_2$CO$_3$ (1.5 wt.%)-Na$_2$MoO$_4$ (8.0 wt.%) melt, temperature 1123 K, current density 5 mA·cm$^{-2}$.

### 3.3. Microstructure of Molybdenum Plates before and after Electrodeposition

The roughness ($R_a$–the average profile arithmetic deviation) of the initial molybdenum substrates measured on a profilometer was 0.06–0.15 μm. The roughness values of the substrates obtained by AFM (see Table 1) are consistent with the values obtained by a standardized method using a profilometer.

However, they cannot completely correspond to each other as calculated, without taking into account the baseline length. However, these data are important because they reflect the surface relief at a more subtle level.

The data in Table 1 show that the parameters of surface roughness are very different depending on the measured area, and the main contribution to the roughness values introduces the transverse direction. The microstructure of the molybdenum plate cross-section after etching is shown in Figure 3.

**Table 1.** Roughness values obtained using an atomic force microscope (AFM).

| Sample | Measurement Area, µm | Roughness Values, µm | | By Area |
|--------|----------------------|----------------------|--------------------|---------|
| | | By Line | | |
| | | Along the Sample | Across the Sample | |
| Substrate | 80 × 80 | 0.13 | 0.73 | 0.75 |
| | 20 × 20 | 0.06 | 0.27 | 0.27 |
| | 5 × 5 | 0.01 | 0.1 | 0.1 |
| Coated | 80 × 80 | 0.31 | 0.19 | 0.28 |
| | 20 × 20 | 0.25 | 0.18 | 0.2 |
| | 5 × 5 | 0.12 | 0.14 | 0.15 |

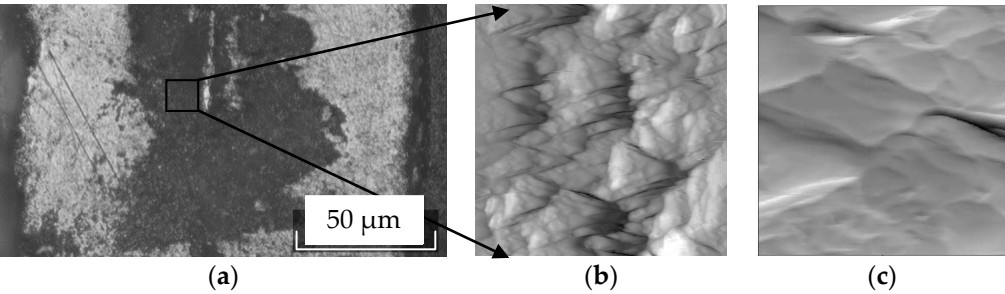

| (a) | (b) | (c) |
|-----|-----|-----|

**Figure 3.** Microimage of the original molybdenum plate cross-section after etching: (**a**) optical microscope, (**b**) AFM area 5.93 µm × 5. 93 µm, (**c**) AFM area 1.14 µm × 1.14 µm.

As can be seen from the images obtained using an optical and atomic force microscope, the initial molybdenum substrates had a rolled structure with longitudinal deformation bands (see Table 2).

**Table 2.** Micrographs of a flat molybdenum plate before and after the electrochemical synthesis of the Mo$_2$C coating obtained on optical and atomic force microscopes.

| Source | Substrate | Coated Sample |
|--------|-----------|---------------|

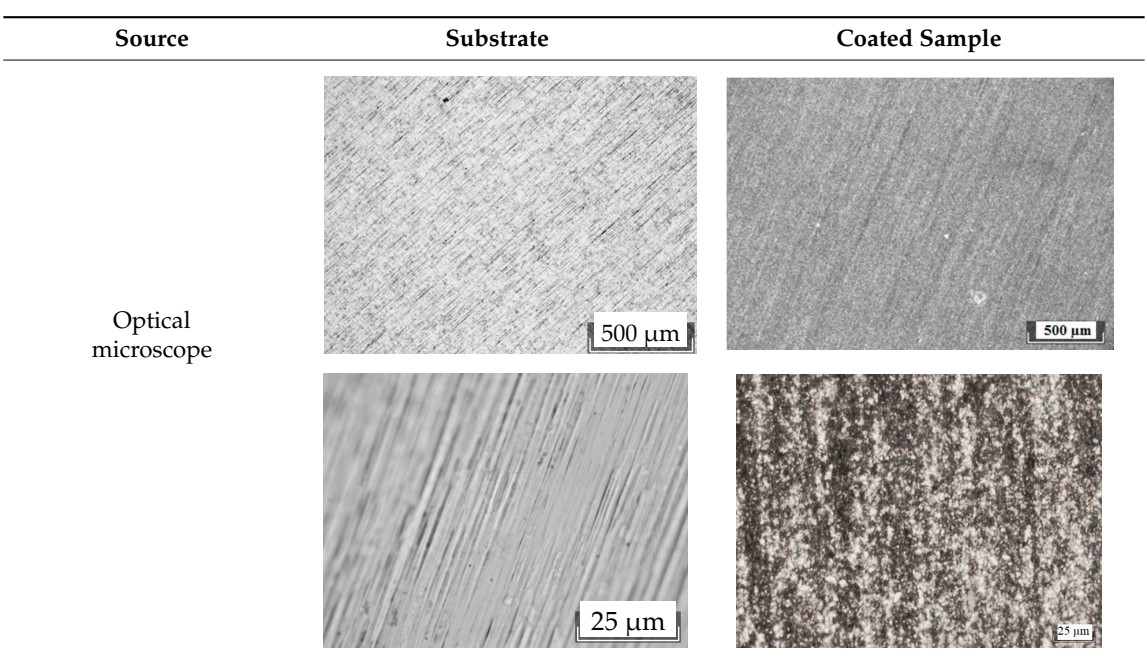

**Table 2.** *Cont.*

| Source | Substrate | Coated Sample |
|---|---|---|
| AFM (scanning area, μm²) | | |

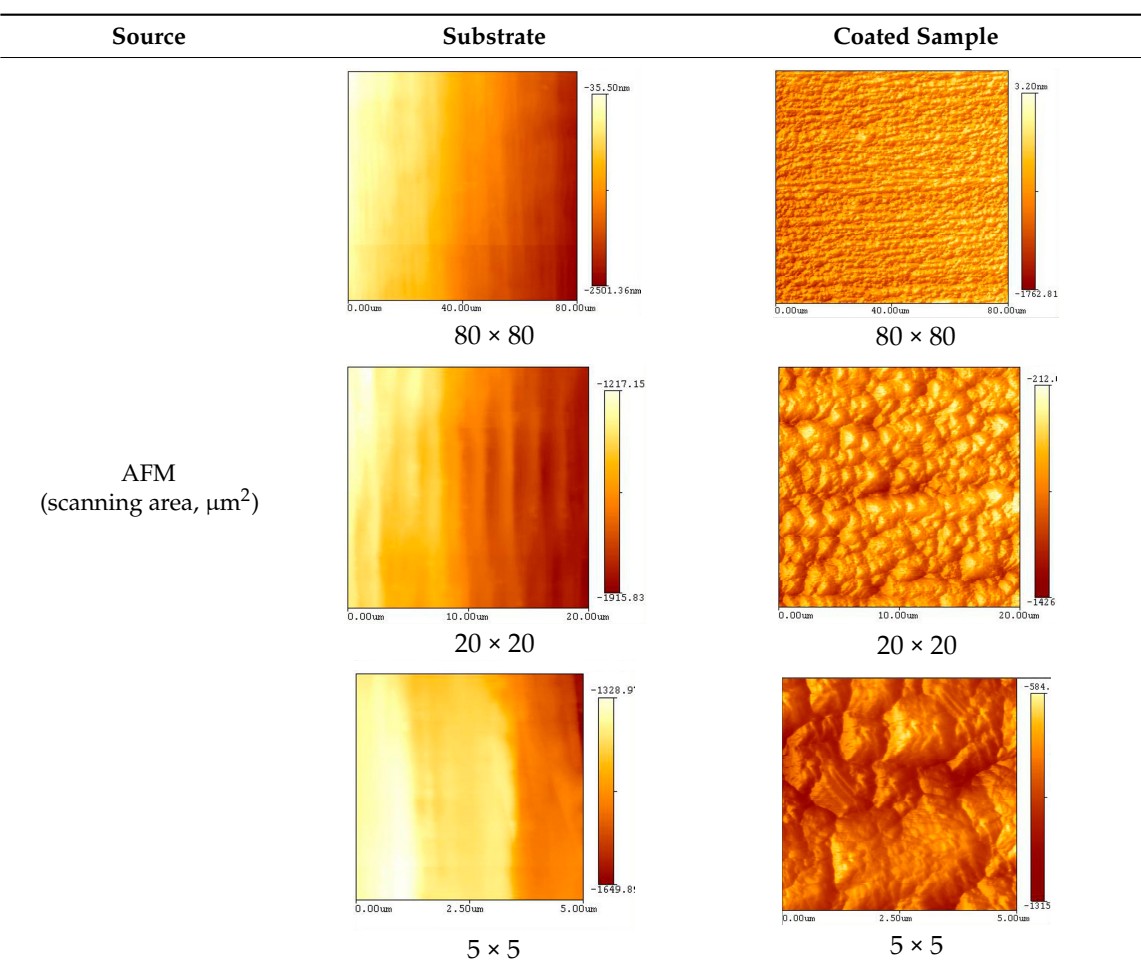

The roughness ($R_a$) of the Mo$_2$C coatings on molybdenum, determined using a profilometer, was 0.06–0.35 μm. After electroplating, the molybdenum semi-carbide coating had roughness values of the same order in different directions, but the longitudinal profile of the sample was already becoming more prominent than the transverse one (Table 1). This was due to the "overgrowth" cathodic deposit rolled strips and forming a coarse deposit in the longitudinal direction (Figure 4).

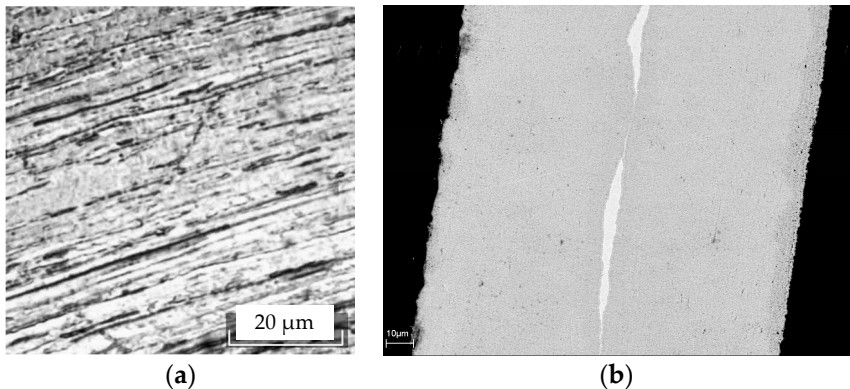

(**a**)　　　　　　　　　　　　　　(**b**)

**Figure 4.** Microimages of a flat molybdenum plate cross-section: (**a**) an original substrate after cross-section preparation and its etching; (**b**) after electrolysis.

After preparing cross-section, a molybdenum substrate was often delaminated (Figure 4a). After electrolysis molybdenum plate thickness increased by 40 microns. On a cross-section, the fine-dispersed type homogeneous structure was observed with residual located in the center of the thin section in the form of initial molybdenum elongated "islands" (Figure 4b). The width of the "islands" did not exceed 5 μm and the presence of a diffusion zone between the substrate and the coating was not established.

### 3.4. Microstructure of Goffered Molybdenum Plates before and after Electrodeposition

It should be noted that during preparation of the cross-section, the goffered plate did not stratify, which is associated with deformation of the obtained plate during corrugation (Figure 5a). In the case of goffered samples, a radical change in the structure was also observed, since prior to electrodeposition the structural elements were predominantly longitudinal (rolled strip, less than 1 μm thick), then after the synthesis of the coating it became transverse (Figure 5b).

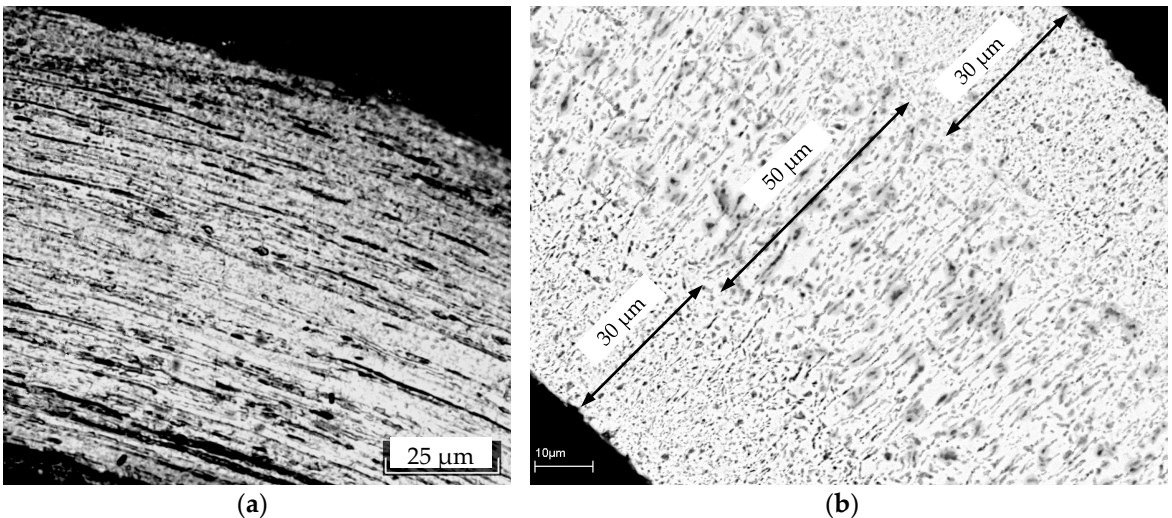

(**a**)  (**b**)

**Figure 5.** Microimages of a goffered molybdenum plate cross-section: (**a**) an original substrate after cross-section preparation and its etching; (**b**) after electrolysis.

The cross-section consists of two external zones (30 μm + 30 μm) and an internal zone 50 μm thick (Figure 5b). XRD of goffered and crushed plates shows the presence of only one phase of molybdenum semi-carbide, and therefore the zones differ only in structure. Full carburization of the goffered plate was confirmed by microanalysis. After deposition of the coating, the sample thickness became 110 μm.

### 3.5. Microstructure of Molybdenum Wire before and after Electrodeposition

The microstructure of the cross-section of the original 250 μm thick molybdenum wire is shown in Figure 6a. After electrolysis, the thickness of the samples reached 270–280 μm, i.e., the increase is 20–30 μm. The central zone with a diameter of 160 μm did not change its structure and the external zone with dimensions of 60–70 μm was radially directed and similar to the structures of the flat sample and the outer zones of the goffered plate (Figure 6b). On a thin section, a small layer several microns thick was also observed at the interface of molybdenum/molybdenum carbide, which has a finely dispersed structure.

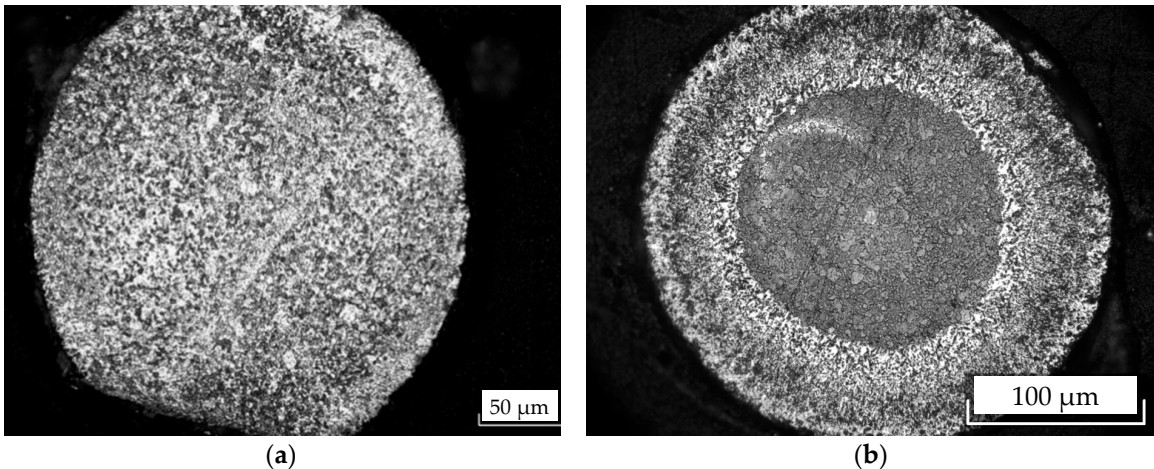

(**a**)                                                    (**b**)

**Figure 6.** Microimages of a molybdenum wire cross-section: (**a**) an original substrate after cross-section preparation and its etching; (**b**) after electrolysis.

*3.6. Molybdenum Substrate Behavior during $Mo_2C$ Electrochemical Synthesis and Its Impact on the Deposit Microstucture*

A common feature of all molybdenum products is the reduction of the substrate thickness during the electrochemical synthesis of $Mo_2C$. Moreover, an increase of the synthesis time led to a greater decrease of molybdenum substrate thickness. One explanation for this decrease may be the interaction of molybdenum with electrolyte used for electrochemical synthesis. Indeed, $Mo_2C$ coating is a porous layer (see Figure 7), which could not prevent a metal–salt reaction.

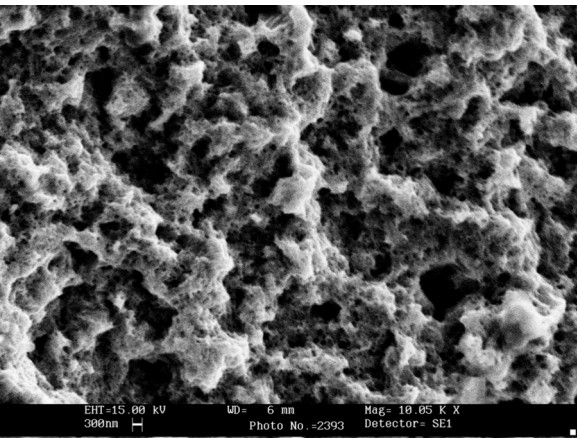

**Figure 7.** The morphology of the $Mo_2C$ coating on a molybdenum substrate obtained from the $NaCl$-$KCl$-$Li_2CO_3$ (1.5 wt.%)-$Na_2MoO_4$ (8.0 wt.%) melt. Temperature 1123 K, current density 5 mA·cm$^{-2}$.

There is an equilibrium known in melts containing molybdate ions [23,24]:

$$2MoO_4^{2-} \leftrightarrow Mo_2O_7^{2-} + O^{2-} \tag{5}$$

In turn, dimolybdate ions can interact with molybdenum by the reaction [25]:

$$2Mo_2O_7^{2-} + Mo \leftrightarrow 3MoO_2 + 2MoO_4^{2-} \tag{6}$$

The aforementioned interaction mechanism is one possible cause of the reduction in substrate thickness, and molybdenum substrate behavior during deposition of $Mo_2C$ coatings from the $NaCl$-$KCl$-$Na_2MoO_4$-$Li_2CO_3$ melt requires further study.

Another important result of this work is the conclusion about the influence of the substrate shape on the cathodic structure of deposit. Thus, the $Mo_2C$ coating on a flat plate is more dense (see Figure 4b), and in a goffered plate (see Figure 5b) and wire (see Figure 6b) carbide layers have a similar morphology and looser cathodic deposits.

Obtained coatings were tested as a catalyst for the water–gas shift reaction:

$$CO + H_2O \leftrightarrow CO_2 + H_2 \tag{7}$$

The steady-state reaction rates for the $Mo_2C/Mo$ coatings were higher than those for the bulk $Mo_2C$ [4] and commercial $Cu/ZnO/Al_2O_3$ catalysts over the temperature range explored (Figure 8). The catalytic activity is enhanced by at least three orders of magnitude compared to that of the bulk $Mo_2C$ phase if molybdenum carbides are present as a thin submicron layer on a molybdenum substrate. The methanation reaction was completely suppressed in the whole temperature range studied on the $Mo_2C/Mo$ coatings. The catalytic activity remained constant during 500 h on-stream (Figure 9). The coatings were also stable during the thermal cycling, while the activity of commercial catalysts tends to decrease with time.

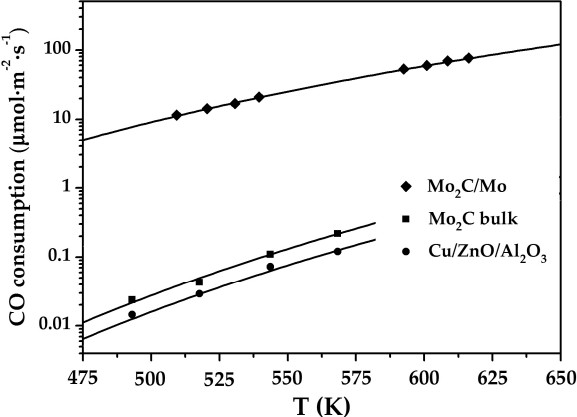

**Figure 8.** Water–gas shift reaction rate as a function of temperature on different catalysts. Reaction conditions: $p_{CO} = 0.003$ atm, $p_{H2O} = 0.013$ atm, $p_{H2} = 0.4$ atm, balance-He. Total flow: 50 cm$^3$·min$^{-1}$ (STP).

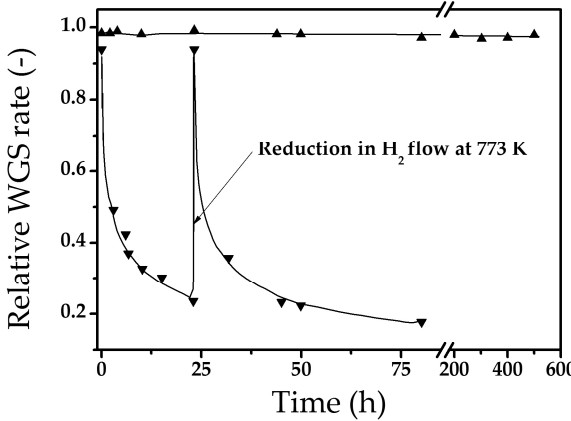

**Figure 9.** Stability of $Mo_2C/Mo$ composition (▲) compared with bulk $Mo_2C$ (▼) under water–gas shift (WGS) reaction conditions (same in Figure 8).

Using articles of different shape with the $Mo_2C$ coatings based on the mathematical model and the thermodynamic parameters calculation, a countercurrent microstructured reactor–heat exchanger was designed [25].

## 4. Conclusions

Molybdenum semi-carbide coatings with a hexagonal crystal lattice on various configurations molybdenum substrates in the NaCl-KCl-Li$_2$CO$_3$ (1.5 wt.%)-Na$_2$MoO$_4$ (8.0 wt.%) melt were obtained.

A mechanism for the interaction of molybdenum with molten salt in the electrochemical synthesis process leading to a decrease in the thickness of molybdenum substrates was proposed.

The effect of the molybdenum substrate shape on the microstructure of the Mo$_2$C coating was studied. It has been established that in the goffered plates and wire, the layers of molybdenum semi-carbide have a similar morphology and looser cathodic deposits in comparison with the Mo$_2$C coating on a flat plate.

It was determined that the molybdenum carbide coatings on a molybdenum substrate (Mo$_2$C/Mo) show a catalytic activity in the water–gas shift reaction by at least three orders of magnitude higher than that of the bulk Mo$_2$C phase. The catalytic activity remained constant during 500 h for the water–gas shift reaction.

**Author Contributions:** Investigation, A.D.; Supervision, S.K.; Visualization, O.M.; Writing–original draft, A.D.; Writing–review & editing, S.K.

**Funding:** This research received no external funding.

**Conflicts of Interest:** The authors declare no conflict of interest.

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
