# Peer review of "Effect of the Molybdenum Substrate Shape on Mo2C Coating Electrodeposition"

_coatings, doi:10.3390/coatings8120442_

Reviewer 1 Report
In this paper, the authors reported on the effect of the molybdenum substrate with different configuration Mo2C coatings.
Specific comments on the paper are listed below:
· The novelty of the work should be explained better. What is new in this paper? What is the importance of this work compared to other published work? The authors should explain better with reference what is new in this paper and the reason why the authors did this work.
· The state of the art is too superficial. The authors should increase the references to more recent papers (2017, 2018, 2019).
· Authors need to explain the listed advantages of the reported technique compared to the other process reported in literature.
· In the abstract, add the obtained results.
· In the section 2.1, what was the used HCl flow in the synthesis?
· In the section 2.1, other information about all the salts should added.
· In the section 2.2, the authors should say what are the characteristics of every used substrate (cas number, code, ...).
· Did the authors evaluate the stability of the synthesized materials? If so, please add some data or consideration about this point.
· The authors should improve the quality of the figures.
· The authors should remove some highlighted words in the text.
· The authors should correct the typos and grammatical errors in the manuscripts.
Author Response
Response to Reviewer 1 Comments
Point 1: The novelty of the work should be explained better. What is new in this paper? What is the importance of this work compared to other published work? The authors should explain better with reference what is new in this paper and the reason why the authors did this work.
·
· Response 1: The novelty of this investigation is study of the influence substrate form on the microstructure and lattice of Mo2C coatings. It was found that in all cases on different form substrates during the high temperature electrochemical synthesis Mo2C coatings only with the hexagonal lattice are formed. Due to this, the catalytical activity of Mo2C produced by electrochemical method by three orders of magnitude higher in comparison with other methods and this is the reason why this work has been done. We improved the aims of our study that to explain better the novelty “The objective of the present work was electrodeposition of molybdenum carbide coatings on different shapes molybdenum substrates in molten salts, its lattice and microstructure characterization.
· This work also presents the development of new generation of highly active and stable catalysts in the form of coatings based on the Mo2C/Mo system obtained by electrochemical synthesis.”
·
Point 2: The state of the art is too superficial. The authors should increase the references to more recent papers (2017, 2018, 2019).
·
Response 2: The authors significantly increase the list of references including the recent papers (2017, 2018). New references are highlighted by yellow.
·
Point 3: Authors need to explain the listed advantages of the reported technique compared to the other process reported in literature.
·
Response 3: Advantages of electrochemical technique are described in the text of revised manuscript (lines 45-55). However, the main advantage of this technique is crystallization of Mo2C coatings only with the hexagonal lattice that provide a very high catalytic activity of the Mo2C/Mo composition.
·
Point 4: In the abstract, add the obtained results.
Response 4: The obtained results were added to the abstract. “The molybdenum carbide coatings on a molybdenum substrate (Mo2C/Mo) show a catalytic activity in the water gas shift (WGS) reaction by at least three orders of magnitude higher than that of the bulk Mo2C phase. The catalytic activity remained constant during 500 hours for the water gas shift reaction”.
·
Point 5: In the section 2.1, what was the used HCl flow in the synthesis?
·
Response 5: The HCl flow was used that to prevent hydrolysis of NaCl and KCl salts. This is a common procedure for preparation of these salts.
·
Point 6: In the section 2.1, other information about all the salts should added.
·
Response 6: Information for all salts was added to the manuscript. “Li2CO3 (Sigma-Aldrich, purum, ≥99 wt. %) and Na2MoO4·H2O (Sigma-Aldrich, purum, ≥99.5 wt. %) were dried for 24 h at 473 K.”
·
Point 7: In the section 2.2, the authors should say what are the characteristics of every used substrate (cas number, code, ...).
·
Response 7: The characteristics of substrates and the surface preparation were added to the text. “Various molybdenum substrates were used: flat plates (Aldrich, 99.9+ wt. %, CAS number 7439-98-7) of size 100 × 10 × 0.1 mm; goffered plates of the same size with the corrugation height of 1.4 mm; wire (Aldrich, wire reel, 50m, diameter 0.25mm, annealed, 99.95% wt. %, CAS number 7439-98-7) 250 μm in diameter. In order to remove the organic impurities from the surface, the molybdenum plates were placed in boiling xylene for 1 h and were then heated in a furnace at 413 K to desorb the xylene that remained on the surface”.
·
Point 8: Did the authors evaluate the stability of the synthesized materials? If so, please add some data or consideration about this point.
·
· Response 8: Data concerning the catalytic activity and stability of the Mo2C/Mo composition were added to the manuscript. “Obtained coatings were tested as a catalyst for the water-gas shift reaction:
(7) |
The steady-state reaction rates for the Mo2C/Mo coatings were higher than those for the bulk Mo2C [4] and commercial Cu/ZnO/Al2O3 catalysts over in the temperature range explored (Figure 8). The catalytic activity is enhanced by at least three orders of magnitude comparing to that of the pure Mo2C phase, if molybdenum carbides are present as a thin submicron layer on a molybdenum substrate. The methanation reaction was completely suppressed in the whole temperature range studied on the Mo2C/Mo coatings. The catalytic activity remained constant during 500 hours on-stream (Figure 9). The coatings were also stable during the thermal cycling, while the activity of commercial catalysts tends to decrease with time.”
·
Point 9: The authors should improve the quality of the figures.
·
·Response 9: The quality of the figures was improved.
·
Point 10: The authors should remove some highlighted words in the text.
·
·Response 10: The highlighted words were removed from the text.
·
Point 11: The authors should correct the typos and grammatical errors in the manuscripts.
·
Response 11: The authors corrected the typos and grammatical errors. From the point of view, the Native speaker the English of the manuscript is satisfactory.
The response in attached file.

Reviewer 2 Report
The manuscript presents some characteristic features of molybdenum carbide coatings electrodeposited on various configurations of Mo substrate (flat plate, goffered plate, wire) from an alkali chloride melt. The electrolyte was NaCl-KCl eutectic, in which Li2CO3 (1.5 wt.%) and Na2MoO4 (8.0 wt.%) as precursors were dissolved. Preparation of coatings was performed at 1123 K temperature for 7 h, using a constant value of cathodic current density, 5 mA·cm−2.
In general, the characterization was almost qualitatively, regarding shape of cyclovoltammograms (linear sweep voltammograms were not presented), XRD patterns (a standard card for hexagonal lattice of Mo2C should be mentioned), roughness measured by AFM (only values for flat plate were reported in a Table), surface morphology (using optical microscope) and cross-section microstructure. Some observations and questions are mentioned below:
· In Abstract: There is not enough information regarding the shape of Mo support and operating parameters of electrodeposition, as composition of melt (% of NaCl, KCl, Li2CO3), temperature, galvanostatic conditions. Cyclic voltammograms recorded comparatively on Mo and glassy carbon, as well as characterization techniques were not mentioned. Usually the abstract should provide a self-standing summary of the performed work.
· In Introduction, some works and their results about Mo2C electrodeposition from molten salts must be cited (Koyama et al, 1984, Topor et al, 1988, Malyshev et al, 2008). A more extensive introduction also discussing more recent papers in the field (i.e. Malyshev et al., Review of the electrodeposition of molybdenum carbide on the surfaces of disperse dielectric and semiconductor materials, Acta Chimica Slovaca, Vol. 5, No. 2, 2012, pp. 139—144; Malyshev et al., Electrodeposition of tungsten and molybdenum carbide onto the surfaces of disperse dielectric and semiconductor materials, Mat.-wiss. u. Werkstofftech. 2014, 45 , No. 1, DOI: 10.1002/mawe.201400189; Stulov et al., Electrochemical Methods for Obtaining Thin Films of the Refractory Metal Carbides in Molten Salts, Int. J. Electrochem. Sci., 12 (2017) 5174 – 5184, doi: 10.20964/2017.06.52; Malyshev et al., High Temperature Ionic Liquids Electrosynthesis of Mo2C Protective Coatings on Diamond, Boron Nitride, Silicon and Boron Carbide, REV.CHIM.(Bucharest), 69, No. 3, 2018, pp.544–548) is suggested.
· Keywords: please replace “articles of different shape” with metallic substrates of different shapes”.
· In Experimental section: which is the surface preparation of Mo substrates before electrodeposition process?
· In Results and Discussion:
- section 3.1, lines 89 and 108: MoO42–are rather ions, not particles.
- the authors claimed in the Experimental section that the electrochemical investigations have been performed on Mo and glassy carbon as working electrodes. However, the presented cyclic voltammograms are only related on the Mo working electrode. The electrochemical investigation showing only two voltammograms at one value of the scan rate is insufficient.
-in a similar manner, the authors showed the results of the Mo2C electrodeposition at a single applied current density. Which is the influence of the applied c.d, of the melt composition, of the applied temperature, on the deposit characteristics? Which is the duration of the electrodeposition process (also to be mentioned in the Experimental section)? Which is the electrodeposition rate against the applied operation parameters and the selected metallic substrate?
- as a part of XRD discussion, a standard card for hexagonal lattice of Mo2C should be mentioned, evidencing the characteristic peaks, etc.
- the authors declared: “ The formation of Mo2C with a hexagonal lattice in the process of electrochemical synthesis occurs due to specific conditions (electric field, double layer, high temperature) of the electrocrystallization process.” How? There is no discussion, only a presentation of the experimental results.
-In Table 1, which kind of roughness is presented, Raverage or RRMS?
-English must be substantially improved, including Abstract. Some sentences (for instance at lines 90,91; line 129; lines 187–189) should be reformulated.
· The authors claimed at the end of Introduction section that: “The objective of the present work was the electrodeposition of molybdenum carbide coatings on different shapes of molybdenum substrates in molten salts and its characterization for catalytic applications”. However, no experimental results are presented related to the use of Mo2C for catalytic applications.
· As a general view, too limited discussion is provided to support all the experimental results.
Author Response
Response to Reviewer 2 Comments
Point 1: Usually the abstract should provide a self-standing summary of the performed work.
Response 1: The obtained results were added to the abstract. “The molybdenum carbide coatings on a molybdenum substrate (Mo2C/Mo) show a catalytic activity in the water gas shift (WGS) reaction by at least three orders of magnitude higher than that of the bulk Mo2C phase. The catalytic activity remained constant during 500 hours for the water gas shift reaction”.
Point 2: -In Introduction, some works and their results about Mo2C electrodeposition from molten salts must be cited (Koyama et al, 1984, Topor et al, 1988, Malyshev et al, 2008). A more extensive introduction also discussing more recent papers in the field (i.e. Malyshev et al., Review of the electrodeposition of molybdenum carbide on the surfaces of disperse dielectric and semiconductor materials, Acta Chimica Slovaca, Vol. 5, No. 2, 2012, pp. 139—144; Malyshev et al., Electrodeposition of tungsten and molybdenum carbide onto the surfaces of disperse dielectric and semiconductor materials, Mat.-wiss. u. Werkstofftech. 2014, 45 , No. 1, DOI: 10.1002/mawe.201400189; Stulov et al., Electrochemical Methods for Obtaining Thin Films of the Refractory Metal Carbides in Molten Salts, Int. J. Electrochem. Sci., 12 (2017) 5174 – 5184, doi: 10.20964/2017.06.52; Malyshev et al., High Temperature Ionic Liquids Electrosynthesis of Mo2C Protective Coatings on Diamond, Boron Nitride, Silicon and Boron Carbide, REV.CHIM.(Bucharest), 69, No. 3, 2018, pp.544-548) is suggested.
Response 2: The authors significantly increase the list of references including papers suggested by Reviewer. Additional references are highlighted by yellow.
Point 3: Keywords: please replace “articles of different shape” with metallic substrates of different shapes”.
Response 3: We did the change of the keyword in accordance with suggestion of Reviewer.
Point 4: In Experimental section: which is the surface preparation of Mo substrates before electrodeposition process?
Response 4: The characteristics of substrates and the surface preparation were added to the text of revised manuscript. “Various molybdenum substrates were used: flat plates (Aldrich, 99.9+ wt. %, CAS number 7439-98-7) of size 100 × 10 × 0.1 mm; goffered plates of the same size with the corrugation height of 1.4 mm; wire (Aldrich, wire reel, 50m, diameter 0.25mm, annealed, 99.95% wt. %, CAS number 7439-98-7) 250 μm in diameter. “In order to remove the organic impurities from the surface, the molybdenum plates were placed in boiling xylene for 1 h and were then heated in a furnace at 413 K to desorb the xylene that remained on the surface”.
Point 5: -In Results and Discussion:
- section 3.1, lines 89 and 108: MoO42 – are rather ions, not particles.
Response 5: The changes have been done in revised version lines 103 and 122.
Point 6: - the authors claimed in the Experimental section that the electrochemical investigations have been performed on Mo and glassy carbon as working electrodes. However, the presented cyclic voltammograms are only related on the Mo working electrode. The electrochemical investigation showing only two voltammograms at one value of the scan rate is insufficient.
Response 6: The authors used glassy carbon electrode for determination the quality of an equimolar mixture NaCl-KCl (concentration of oxide and hydroxide ions in the melt), but these results were not included to the manuscript. Thus, in revised version we indicated only Mo working electrode. Indeed, voltammetric curves are presented at one value of the scan rate. We used different scan rates, but the shape of voltammograms was the same. It is sufficient to use different sweep rates that to use diagnostic criteria of cyclic voltammetry for determination the nature of electrode process (reversible, quasi-reversible, irreversible). However, it was not the task of this study.
Point 7: -in a similar manner, the authors showed the results of the Mo2C electrodeposition at a single applied current density. Which is the influence of the applied c.d, of the melt composition, of the applied temperature, on the deposit characteristics? Which is the duration of the electrodeposition process (also to be mentioned in the Experimental section)? Which is the electrodeposition rate against the applied operation parameters and the selected metallic substrate?
Response 7: We did not study the influence of current density and temperature because such work increases very sharply the volume of microstructure investigations. Synthesis of Mo2C on molybdenum plates and wire was performed at 1123 K for 7 h at the cathodic current density of 5 mA cm-2. The electrodeposition rate was smaller than theoretical one due to the formation of dendrites, which were removed from substrates during washing the deposits.
Point 8: - as a part of XRD discussion, a standard card for hexagonal lattice of Mo2C should be mentioned, evidencing the characteristic peaks, etc.
Response 8: The card for hexagonal lattice of Mo2C and the characteristic peaks (Fig. 2) were indicated in revised version.
Point 9: - the authors declared:
“The formation of Mo2C with a hexagonal lattice in the process of electrochemical synthesis occurs due to specific conditions (electric field, double layer, high temperature) of the electrocrystallization process.” How? There is no discussion, only a presentation of the experimental results.
Response 9: The authors declared this conclusion because in our studies we proposed that electrocrystallization at the cathode is accompanied by an effect similar to the action of high pressure. This assumption was made based on the analysis of structure peculiarities of b-Ta, which is formed during electrolysis; the fact of formation of solid electrolytic solution Nb(O) with extremely high oxygen content (up to 17 at %) without reduction of cubic symmetry; the fact of formation of monoxide with an extremely extended homogeneity region 1.0 << span=""> O/Nb £1.7; and the electrochemical synthesis of Mo2C with a hexagonal lattice in this manuscript.
Point 10: -In Table 1, which kind of roughness is presented, Raverage or RRMS ?
Response 10: The parameter Ra – the average profile arithmetic deviation was used for characterization of roughness.
Point 11: -English must be substantially improved, including Abstract. Some sentences (for instance at lines 90, 91; line 129; lines 187-189) should be reformulated.
Response 11: Abstract and indicated by Reviewer sentences were rewritten (lines 105,106; lines 145,146; lines 202, 203 and 205 in revised manuscript).
Point 12: The authors claimed at the end of Introduction section that: “The objective of the present work was the electrodeposition of molybdenum carbide coatings on different shapes of molybdenum substrates in molten salts and its characterization for catalytic applications.”. However, no experimental results are presented related to the use of Mo2C for catalytic applications.
Response 12: Data concerning the catalytic application and stability the of Mo2C/Mo composition were added to the manuscript. “Obtained coatings were tested as a catalyst for the water-gas shift reaction:
CO + H2O « CO2 + H2 | (7) |
The steady-state reaction rates for the Mo2C/Mo coatings were higher than those for the bulk Mo2C [4] and commercial Cu/ZnO/Al2O3 catalysts over in the temperature range explored (Figure 8). The catalytic activity is enhanced by at least three orders of magnitude comparing to that of the pure Mo2C phase, if molybdenum carbides are present as a thin submicron layer on a molybdenum substrate. The methanation reaction was completely suppressed in the whole temperature range studied on the Mo2C/Mo coatings. The catalytic activity remained constant during 500 hours on-stream (Figure 9). The coatings were also stable during the thermal cycling, while the activity of commercial catalysts tends to decrease with time.”
The response in attached file.

Round 2
Reviewer 1 Report
Accepted
Author Response
Thank you very much for your job.
Reviewer 2 Report
The authors have made the recommended modifications. However:
· Page 3, Figure 1-you have 3 CVs, but you mentioned only Figure1a,b,- please check and correct.
· XRD- the authors claimed they have added XRD card, however it was still not present in the manuscript. Please check.
Author Response
Thank you very much for your job.
Page 3, Figure 1-you have 3 CVs, but you mentioned only Figure1a,b,- please check and correct
-There are only two CVs in the original manuscript (see .docx file). Perhaps there were technical errors during conversion of original .docx file to PDF.
XRD-the authors claimed they have added XRD card, however it was still not present in the manuscript. Please check.
-The XRD card is added, lines 127–128.